# Reinforcement Learning with Foundation Priors: Let the Embodied Agent Efficiently Learn on Its Own

**Weirui Ye**[123]    **Yunsheng Zhang**[23]    **Haoyang Weng**[1]    **Xianfan Gu**[2]    **Shengjie Wang**[123]
**Tong Zhang**[123]    **Mengchen Wang**[1]    **Pieter Abbeel**[4]    **Yang Gao**[123] [*]
[1]Tsinghua University, [2]Shanghai Qi Zhi Institute
[3]Shanghai Artificial Intelligence Laboratory, [4]UC Berkeley

**Abstract:** Reinforcement learning (RL) is a promising approach for solving robotic manipulation tasks. However, it is challenging to apply the RL algorithms directly in the real world. For one thing, RL is data-intensive and typically requires millions of interactions with environments, which are impractical in real scenarios. For another, it is necessary to make heavy engineering efforts to design reward functions manually. To address these issues, we leverage foundation models in this paper. We propose Reinforcement Learning with Foundation Priors (RLFP) to utilize guidance and feedback from policy, value, and success-reward foundation models. Within this framework, we introduce the Foundation-guided Actor-Critic (FAC) algorithm, which enables embodied agents to explore more efficiently with automatic reward functions. The benefits of our framework are threefold: (1) *sample efficient*; (2) *minimal and effective reward engineering*; (3) *agnostic to foundation model forms and robust to noisy priors*. Our method achieves remarkable performances in various manipulation tasks on both real robots and in simulation. Across 5 dexterous tasks with real robots, FAC achieves an average success rate of 86% after one hour of real-time learning. Across 8 tasks in the simulated Meta-world, FAC achieves 100% success rates in 7/8 tasks under less than 100k frames (about 1-hour training), outperforming baseline methods with manual-designed rewards in 1M frames. We believe the RLFP framework can enable future robots to explore and learn autonomously in the physical world for more tasks. Visualizations and code are available at https://yewr.github.io/rlfp.

**Keywords:** Reinforcement Learning, Foundation Models, Robotics

## 1   Introduction

Reinforcement Learning (RL) has achieved remarkable success in various domains, including video games [1, 2, 3], simulated robotics [4], and embodied agents for decades, such as real robots [5, 6, 7, 8, 9]. However, current RL algorithms encounter two primary challenges: sample efficiency and heavy reward engineering, which hinder deployment in the real world. For example, researchers [1, 10] require millions of data to master games or solve simulated robotics tasks [11, 12, 13]. Such amounts of data are unaffordable in real. And they also depend on manually designed rewards, which is overly burdensome. Therefore, it is essential to address these two issues.

Humans acquire skills through minimal interactions with the environment by leveraging the innate abilities and abundant commonsense accumulated in daily life. They start from reasonable behaviors with fewer aimless explorations, make adjustments and corrections, and reinforce successful behaviors. We take the baby pressing bottom as an example to illustrate this learning paradigm, which is commonly used in psychology [14, 15]. As shown in Fig. 1, a baby named Alice is presented with a novel toy box with a button. The baby observes an adult pressing the button, causing the box to light

---

[*]Corresponding author. Emails: ywr20@mails.tsinghua.edu.cn; gaoyangiiis@tsinghua.edu.cn.

8th Conference on Robot Learning (CoRL 2024), Munich, Germany.

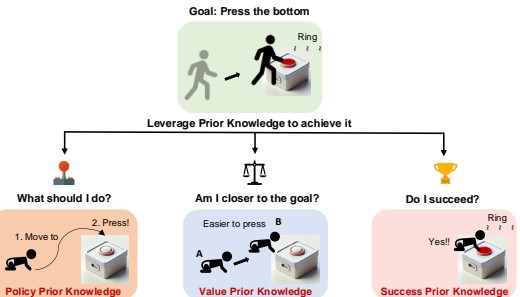

Figure 1: An example of how human solves tasks under the policy, value, and success-reward prior knowledge. The proposed **Reinforcement Learning from Foundation Priors** framework utilizes the corresponding foundation models to acquire prior knowledge.

and make sounds. Alice knows the behavior to solve the task, e.g., pressing the button to activate the toy. She realizes that she can make it easier by positioning the hand closer to the button. Once she sees the toy box light up and hears the sound, she will reinforce the successful trial and repeat it.

Inspired by such a learning paradigm, we explore various foundation models to provide learning signals and enhance learning efficiency. Then two fundamental challenges arise: (1) what is the concrete form to present prior knowledge for RL; (2) how to leverage the corresponding prior knowledge effectively for downstream tasks. Based on the example shown in Fig. 1, three kinds of prior knowledge answer the question: what should I do now? am I closer to the goal? did I succeed? These prior knowledge are aligned with the core concepts well in the Markov Decision Process (**MDP**), namely the policy function, the value function, and the success-reward function. Fortunately, the great success of the foundation models in natural language processing [16, 17, 18, 19] and computer vision [20, 21, 22, 23] makes it possible to acquire considerable and informative prior knowledge.

Consequently, for systematically utilizing the abundant prior knowledge to facilitate the embodied agent efficiently learning on its own, we introduce the Reinforcement Learning from Foundation Priors (**RLFP**) to leverage policy, value, and success-reward prior knowledge. The policy prior gives a warm start behavior of the agent; the value prior informs to reach better states, and the success-reward prior gives the final success feedback.

To verify the efficacy of RLFP, we instantiate an actor-critic algorithm, named Foundation-guided Actor-Critic (**FAC**), inspired by some works about building foundation models [24, 25, 26]. We conduct experiments on real robots and in simulation, and extensive ablations are made in simulation. RLFP demonstrates three key benefits: (1) *Sample efficient learning*. Across the 5 tasks on real robots, FAC can achieve 86% success rates after 1 hour of real-time learning. Across the 8 tasks in the simulated Meta-world, FAC can achieve 100% success rates in 7/8 tasks under less than 100k frames (about 1 hour of training). It surpasses baseline methods that rely on manually designed rewards over 1M frames. (2) *Minimal and effective reward engineering*. The reward function is derived from the value and success-reward prior knowledge, eliminating the need for human-specified dense rewards or teleoperated demonstrations. (3) *Agnostic to prior foundation model forms and robust against noisy priors*. FAC demonstrates resilience under quantization errors in simulations. To ensure high efficiency and performance, we utilize several well-trained foundation models or fine-tuning foundation models. The contributions are:

- We propose the Reinforcement Learning with Foundation Priors (RLFP) framework. It systematically introduces three priors that are essential to embodied agents, and suggests how to leverage the existing foundation models as the priors.

- We propose the Foundation-guided Actor-Critic (FAC), an RL algorithm under the RLFP framework that utilizes the policy, value, and success-reward prior knowledge.

- Empirical results demonstrate the remarkable performances of FAC. The ablations underscore the importance of each priors and validate the robustness against prior qualities.

## 2   Related Work

**Foundation Models for Policy Learning.** The ability to leverage generalized knowledge from large and varied datasets has been proved in the fields of CV and NLP. In embodied AI, researchers attempt to learn universal policies based on large language models (LLMs) or vision-language models

(VLMs). Approaches include training large transformers through imitation learning [27, 28, 29, 30], offline RL [31], or fine-tuning pre-trained VLMs for downstream tasks [8]. Others use LLMs or VLMs for reasoning and control based on language descriptions [32, 33, 34, 35, 36, 37, 38, 39]. These methods rely on human teleoperation for data collection, but scaling this is difficult. Some works generate code policies [40, 41] or predict future videos for action generation [25, 42], though they struggle with robustness due to limited interaction with environments.

**Foundation Models for Representation Learning.** In addition to learning policies from foundation models, some researchers focus on extracting universal representations for downstream tasks. Several works leverage pre-trained visual representations to initialize perception encoders or extract latent image states [43, 44, 45, 46], while others use pre-trained LLMs or VLMs for linguistic instruction encoding [47, 48, 49]. Researchers have also explored applying LLMs/VLMs for universal reward or value representation in RL, such as language-conditioned reward models [50, 51, 52] and universal goal-conditioned value functions trained on large-scale videos [24]. Some approaches acquire value and success signals from human demonstrations [53, 54], while others generate rewards based on value differences to avoid manual reward design [55, 8]. However, these methods face limitations, including inefficient exploration due to a lack of policy-side guidance and unstable optimization from continuous reward signals with high error. Our framework addresses these issues by foundation priors, orthogonal to existing data-efficient RL algorithms [56, 11, 2, 57, 58, 59].

## 3 Method

This paper aims to enable embodied agents to efficiently learn various tasks autonomously. In Sec. 3.1, we formulate the proposed Reinforcement Learning with Foundation Priors (**RLFP**) framework. Next, in Sec. 3.2, we present the Foundation-guided Actor-Critic (**FAC**) algorithm based on RLFP framework. Finally, Sec. 3.3 explains how to obtain the foundation models used in FAC.

### 3.1 Reinforcement Learning with Foundation Priors

We model the tasks for embodied agents as the Goal-Conditioned Markov Decision Processes (GCMDP) $\mathcal{G}$: $\mathcal{G} = (\mathcal{S}, \mathcal{A}, \mathcal{P}, \mathcal{R}, \mathcal{T}, \gamma)$. $\mathcal{S} \in \mathbb{R}^m$ denotes the state. $\mathcal{A}$ is the action space, which is the continuous delta movement of the end effector in this work. $\mathcal{P}$ is the transition probability function. $\mathcal{T}$ is the task identifier. $\mathcal{R}$ denotes the rewards. $\gamma$ is the discounting factor, equal to 0.99 in the work. To learn efficiently and automatically, we propose the Reinforcement Learning from Foundation Priors (**RLFP**) framework by leveraging the policy, value, and success-reward priors.

Here we demonstrate how we formulate the priors in RLFP. Back to the case of Alice in Fig. 1, the commonsense of behavior can be formulated as a goal-conditioned policy function, $M_\pi(s, \mathcal{T}) : \mathcal{S} \times \mathcal{T} \rightarrow \mathcal{A}$. The prior knowledge that the state closer to the button is closer to success can be formulated as the value function $M_\mathcal{V}(s, \mathcal{T}) : \mathcal{S} \times \mathcal{T} \rightarrow \mathbb{R}^1$. The ability to recognize the success state can be formulated as the 0-1 success-reward function $M_\mathcal{R}(s, \mathcal{T}) : \mathcal{S} \times \mathcal{T} \rightarrow \{0, 1\}$, which equals 1 only if the task succeeds. We assume the success-reward prior is relatively precise, given the simplicity of binary classification in determining success. The value and policy prior knowledge are noisier. The RLFP framework is to solve an expansion of $\mathcal{G}$, termed $\mathcal{G}' = (\mathcal{G}, \mathcal{M})$, where $\mathcal{M}$ is the foundation model set that represents various foundation prior knowledge. Here, $M_\pi, M_\mathcal{V}, M_\mathcal{R} \in \mathcal{M}$.

Compared to vanilla RL, all the signals for the RLFP come from the foundation models. The vanilla RL relies on uninformative trial and error explorations and manually designed reward functions. It is not only of poor sample efficiency but also requires much human reward engineering. Instead, in RLFP, prior knowledge from the foundation model set $\mathcal{M}$ provides guidance or feedback on policy, value, and success-reward, enabling more automatic and effective task resolution.

### 3.2 Foudation-guided Actor-Critic

Under the proposed RLFP framework, we instantiate an actor-critic algorithm named Foundation-guided Actor-Critic (FAC), demonstrating how to inject the three priors into RL algorithms.

**Guided by Success-reward Signals.** We consider the task as MDP $\mathcal{G}_1$ with 0-1 success rewards, where $\mathcal{R}_{\mathcal{G}_1} = M_{\mathcal{R}}(s, \mathcal{T}) \in \{0, 1\}$. Inspired by how humans learn from successful trials, we propose a success buffer to store the "successful" trajectories identified by $M_{\mathcal{R}}$. Each time the actor $\pi_\phi$ updates via policy gradient, it also imitates samples from the success buffer $\mathcal{D}_{\text{succ}}$ (if available). The objective is $\mathcal{L}_{\text{succ}}(\phi) = \mathbf{KL}(\pi_\phi(s_t), \mathcal{N}(a_t, \hat{\sigma}^2)), s_t, a_t \sim \mathcal{D}_{\text{succ}}$, where $\hat{\sigma}$ is the standard deviation.

**Guided by Policy Regularization.** To encourage efficient explorations, we regularize the actor $\pi_\phi$ by the policy prior from $M_\pi(s, \mathcal{T})$. Assuming the prior follows Gaussian distributions, the regularization term is $\mathcal{L}_{\text{reg}}(\phi) = \text{KL}(\pi_\phi, \mathcal{N}(M_\pi(s_t, \mathcal{T}), \hat{\sigma}^2))$, which is commonly used in other algorithms [60, 61]. The bias introduced by the policy prior is bounded, shown in Theorem 2.

**Guided by Reward-shaping from Value Prior.** Noisy policy prior can mislead agents to undesirable states, so we propose using the value model $M_{\mathcal{V}}$ to guide exploration and avoid unpromising states. While initializing and fine-tuning with $M_{\mathcal{V}}(s, \mathcal{T})$ is a natural approach, it suffers from catastrophic forgetting. Instead, we employ the **reward-shaping** technique [62] using the potential-based function $F(s, s', \mathcal{T}) = \gamma M_{\mathcal{V}}(s', \mathcal{T}) - M_{\mathcal{V}}(s, \mathcal{T})$, where $\gamma$ is the discount factor. Since $M_{\mathcal{V}}$ estimates state values, $F$ measures the value increase

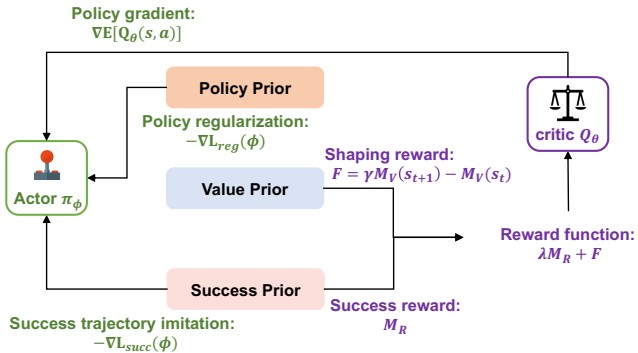

Figure 2: **The overview of Foundation-guided Actor-Critic.** In FAC, rewards are derived from foundation success rewards and value shaping. Besides policy gradient updates, the actor is trained using prior policy regularization and success trajectory imitation.

from $s$ to $s'$. This shaping reward is positive when $s'$ is better than $s$ and shares the same optimal solution as the 0-1 success-reward MDP $\mathcal{G}_1$. Proof and details are in App. A.1.

**Foundation-guided Actor-Critic.** In summary, we deal with a new MDP $\mathcal{G}_2$, where $\mathcal{R}_{\mathcal{G}_2} = \lambda M_{\mathcal{R}} + F$, with $\lambda$ (set to 100) emphasizing success feedback. We train the agent using DrQ-v2 [11], a variant of Actor-Critic, and call the proposed method Foundation-guided Actor-Critic (**FAC**). As shown in Fig. 2, FAC leverages foundation policy guidance and an automatic reward function, enabling the agent to efficiently learn from abundant prior knowledge. The objectives of FAC are detailed in Eq. (1), where tradeoff parameters $\alpha$ and $\beta$ are both set to 1, $y$ is the n-step TD target, and $Q_{\bar{\theta}}$ is the target network. We use clipped double Q-learning [63] to reduce overestimation.

$$\mathcal{L}_{\text{actor}}(\phi) = -\mathbb{E}_{s_t \sim \mathcal{D}}\left[\min_{k=1,2} Q_{\theta_k}(s_t, a_t)\right] + \alpha \mathcal{L}_{\text{succ}} + \beta \mathcal{L}_{\text{reg}}; a_t \sim \pi_\phi(s_t)$$

$$\mathcal{L}_{\text{critic}}(\theta) = \mathbb{E}_{s_t \sim \mathcal{D}}\left[(Q_{\theta_k}(s_t, a_t) - y)^2\right]; y = \sum_{i=0}^{n-1} \gamma^i r_{t+i} + \gamma^n \min_{k=1,2} Q_{\bar{\theta}_k}(s_{t+n}, a_{t+n})$$

$$(1)$$

### 3.3 Acquiring Foundation Prior in FAC.

We focus on leveraging Foundation Priors in RL, not building large-scale foundation models, though we consider it an exciting future direction. For strong performance, we recommend using well-trained or pre-trained models. In this work, we use existing models as proxy foundation models. GPT-4V serves as the success-reward model $M_{\mathcal{R}}$ for real tasks, but due to its poor performance in simulations, we use ground-truth 0-1 success rewards in sim and distill a noisy success reward function for ablations. For value prior, we use the VIP model [24], which predicts value based on current and goal image observations. The policy prior $M_\pi$ is built using code generation [40, 41] or video diffusion models [25]. Details are in the next section and in App. A.3.1 and App. A.3.2.

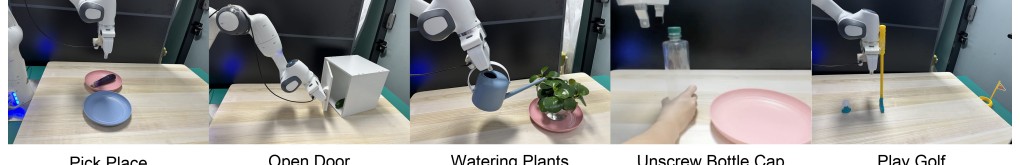

| Pick Place | Open Door | Watering Plants | Unscrew Bottle Cap | Play Golf |

Figure 3: Five tasks on real robots, demonstrating the efficiency and accuracy of FAC in real.

## 4 Experiments

In this section, we provide detailed evaluations of the Foundation-guided Actor-Critic (FAC) on robotics manipulation tasks in both real-world and simulated environments. We also examine the impact of foundation prior knowledge through ablations in simulation, focusing on sample efficiency and robustness. Our experiments aim to answer the following questions: **(a)** How sample-efficient is FAC (Sec. 4.1 and Sec. 4.2); **(b)** What is the significance of each foundation prior (Sec. 4.3); **(c)** How does the quality of the foundation model affect FAC's performance (Sec. 4.3). Additional ablations and running time analysis are in App. A.5. Demo videos are included in the supplementary.

### 4.1 Manipulation tasks on Real Robots

**Experimental Setup.** We set up a real-world tabletop environment using a Franka Emika Panda robot with a 7-DoF arm and a 1-DoF parallel jaw gripper. Observations are collected from RGB images captured by a fixed external camera and a wrist-mounted camera. We designed five dexterous manipulation tasks to evaluate FAC, shown in Fig. 3. The agent learns for one hour per task, except for "Pick Place," which trains for 30 minutes. The number of real-world trajectories collected for Pick Place, Open Door, Watering Plants, Unscrew Bottle Cap, and Play Golf are 40, 75, 60, 50, and 115, respectively. These tasks highlight the need for leveraging prior knowledge for accurate manipulation, which can be challenging for standard RL. The Franka Arm's starting position is fixed for each task, but object positions vary within a predefined range. We evaluate each task across 10 trajectories with slight environment variations. Additional training details are in App. A.2.

**Acquiring Foundation Prior.** (1) Prior Success-Reward: We use GPT-4V to determine success-rewards, evaluating only the final observation of each task. In a test of 20 trajectories per task, GPT-4V demonstrated high accuracy, with no false negatives or positives, except in the Watering Plants task, which had a 25% false-positive rate due to the challenge of correctly orienting the spout. Overall, GPT-4V is an effective proxy for success-reward prior knowledge. (2) Prior Policy: We use GPT-4V to generate code for initial states, building on previous work [40, 41, 34] by providing primitive low-level skills like move_to, grasp, release, and rotate_anticlockwise. The generated code, based on the robot's state, outputs prior actions, allowing us to apply KL loss between the policy and prior actions in non-initial states. (3) Prior Value: We use the VIP model [24], a universal value model trained on large-scale datasets. By inputting goal and current observations, we infer the value of states. Detailed prompts, generated policies, and prior settings are in App. A.3.1.

**Efficient and Safe Exploration on Real Robots.** To achieve higher sample efficiency on real robots, we choose high update-to-data (UTD) ratios with layer normalization in all MLP layers [64]. To achieve safer exploration, we introduce two key modifications. Firstly, we warm up the learning by taking the prior action to gather the first 10 trajectories. Secondly, under the guidance of critics, we selectively choose the superior action $a^* = a_i$ from either the actor $\pi_\phi$ or the policy prior $M_\pi$, where $i = \arg\max_{i \in 1,2} \left[ \min_k Q_{\theta_k}(s, a_i) \right], a_1 \sim \pi_\phi, a_2 \sim M_\pi$. As training progresses, the agent increasingly opts for the actor's actions over the prior, shown in Fig. 4. For the actor, we maintain a constant standard deviation of 0.1.

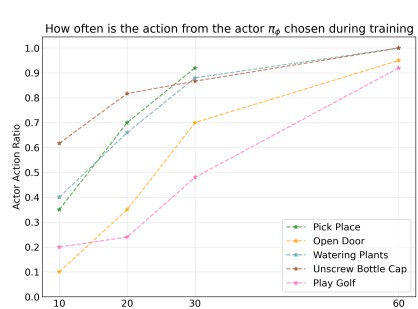

Figure 4: During training, the agent progressively favors actions from the actor, reducing reliance on the prior policy.

**Baselines and Performance Analysis in Real.** We compared our method to two baselines on real robots: (1) Vanilla RL using DrQ-v2 [11] with manually designed rewards, and (2) Policy Prior using the GPT-4V-generated code policy. Results are shown in Tab. 1. Vanilla RL failed in all tasks

Table 1: **Quantitative Results of FAC during One-Hour Learning on the Franka Arm.** The reported results are based on 10 evaluation trials. FAC achieves 86% success rate on average after one hour of training. This performance notably surpasses the code policy prior, underscoring its efficiency.

| Task (Succ.) | Pick Place | Open Door | Watering. | Unscrew. | Play Golfs | Avg. |
|---|---|---|---|---|---|---|
| **Vanilla RL** | 0.00 | 0.00 | 0.00 | 0.00 | 0.00 | 0.00 |
| **Code Policy** | 0.30 | 0.10 | 0.30 | 0.20 | 0.20 | 0.22 |
| **FAC** | 1.00 | 0.90 | 0.80 | 0.90 | 0.70 | **0.86** |

after one hour of training due to excessive exploration. The code policy prior achieved an average success rate of 22%, benefiting from VLM's commonsense knowledge. Notably, FAC, guided by foundation priors, achieved an impressive average success rate of 86% across all tasks.

The learned policy in FAC adapts and refines its approach based on the prior policy to successfully complete tasks, as shown in Fig. 5. While the prior policy struggles with grasping the door handle, FAC persistently secures the handle before pulling, significantly improving performance. FAC's impressive results demonstrate the efficiency of our framework and highlight the potential of leveraging abundant prior knowledge for embodied agents using the RLFP framework.

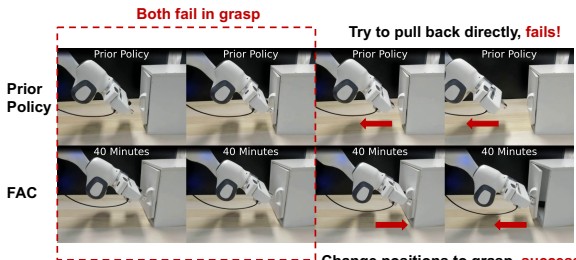

Figure 5: Prior policy attempts to open the door without a successful grasp, whereas FAC persistently tries to secure the handle before pulling back the arm.

## 4.2 Manipulation tasks in Simulations

**Environments.** We conduct experiments in 8 tasks from simulated robotics environment Meta-World [65], widely recognized for their ability to test diverse manipulation skills [60]. We average the success rates over 10 evaluation episodes across 3 runs with different seeds.

**Acquiring Foundation Prior.** (1) Prior Success-Reward: Few foundation models can distinguish success behaviors in embodied AI, and GPT-4V often fails with simulation images. Therefore, we use the ground-truth 0-1 success reward in simulation and distill a noisy success-reward model for ablation in Sec. 4.3. (2) Prior Policy: To show that the prior model can be agnostic to form, we use a diffusion-based policy prior, following the UniPi [25] pipeline, which generates videos using diffusion models and infers actions from an inverse dynamics model. For efficiency, we offline distill a policy model from videos generated by the open-source Seer [26] model, which predicts videos conditioned on images and language instructions. While in-domain fine-tuning is ideally unnecessary, current models fail in the simulator. Thus, we fine-tuned Seer with 10 example videos per task, though these videos are much noisier than the 200k used by UniPi, as shown in the supplementary materials. (3) Prior Value: We use the VIP model [24] with the same setup as in the real robot experiments. Implementation details of the diffusion policy are in App. A.3.2.

**Baselines.** We benchmark our method against the following baselines: **(1)** Vanilla DrQ-v2 [11], with manually designed rewards from the suite; **(2)** R3M [46], VIP [24], where we integrate the DrQ-v2 with either the R3M or VIP visual representation backbones. These baselines also rely on manually designed rewards from the suite; **(3)** UniPi [25]; **(4)** The distilled policy from UniPi.

**Performance Analysis in Meta-World.** We compared our method to baselines on 8 Meta-World tasks with 1M frames. FAC achieved 100% success rates in all tasks, with 7/8 requiring fewer than 100k frames (about 1 hour of training), and the harder bin-picking task needing under 400k frames. In contrast, baseline methods fail to reach 100% success on most tasks. As shown in Fig. 6, FAC significantly outperforms baselines in both sample efficiency and success rates. While DrQ-v2 can complete some tasks, it learns much slower than FAC. Additionally, we warm up the actor with 10 success demos from the diffusion-based prior policy before training. Although the improvement in DrQ-v2 with warmup is noticeable (Fig. 7), it falls behind FAC, which benefits from policy prior.

R3M and VIP backbones provide visual representation prior knowledge for RL, but perform worse than DrQ-v2, likely due to the loss of plasticity in pre-trained models [66]. As UniPi and the dis-

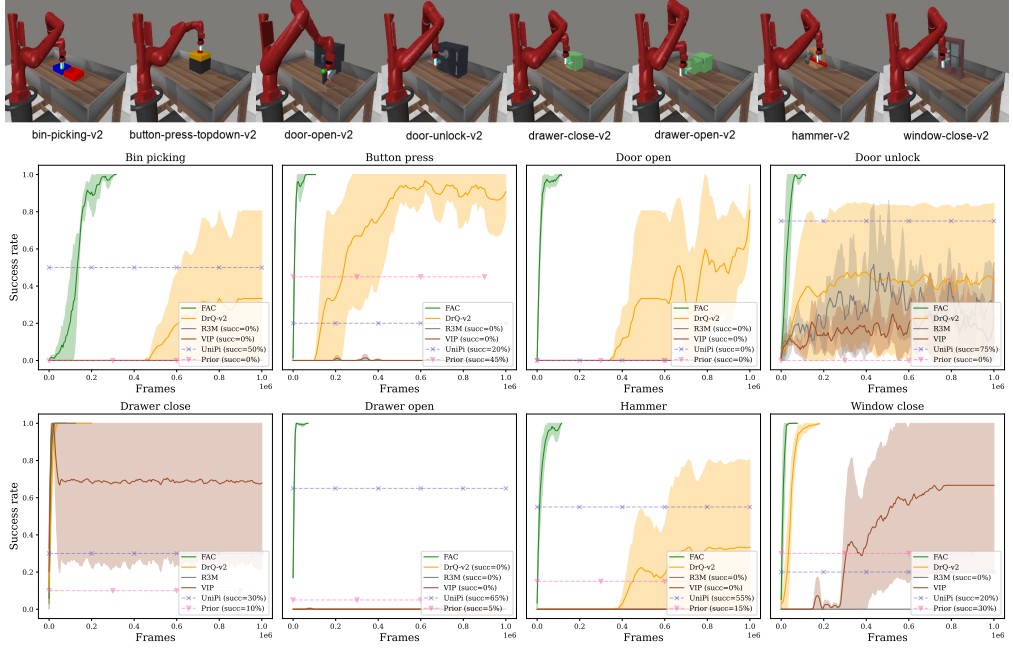

Figure 6: **Success Rate Curves for the 8 Tasks in Meta-World.** Our method consistently achieves **100% success rates** across all tasks, under the constrained performance of the policy prior model. It significantly outperforms the baselines with manual-designed rewards.

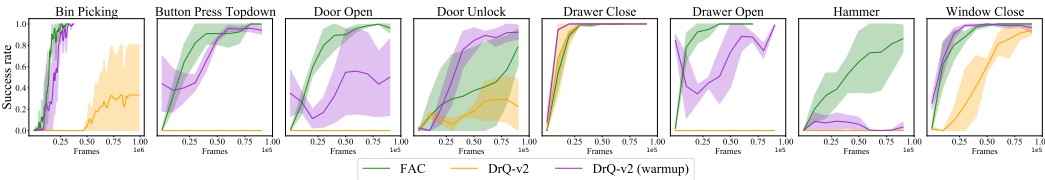

Figure 7: Comparison to the DrQ-v2 (warmup), which warm up the actor by 10 collected success demos from the prior policy model. FAC achieves better performance across the 8 tasks generally.

tilled foundation prior policy lack environment interaction during training, they are represented as horizontal lines in Fig. 6. While UniPi outperforms the distilled prior in most environments, since the latter is learned from UniPi, both are still far inferior to FAC.

## 4.3 Ablation Study in Simulations

In this section, we answer the following questions: (I) What's the importance of the three proposed priors? (II) How does FAC perform with noisier priors?

**Ablation of Each Foundation Prior.** To assess the importance of each foundation prior, we conducted experiments by removing them individually and comparing the results to the full method, along with an ablation study on the success buffer (Fig. 8 (a)). We found the reward prior to be the most crucial. Without it, performance drops significantly or fails, as the reward function reduces to shaping rewards, making policies indistinguishable. Without the policy prior, the agent struggles with difficult tasks like bin-picking and door-opening, and convergence slows on most tasks. Removing the value prior or success buffer reduces sample efficiency, especially for bin-picking, as the value prior helps guide policies under noisy policy priors. Success trajectory imitation further improves sample efficiency. Overall, using all foundation priors yields the best results.

**FAC with Various Quality of Foundation Priors.** As previous ablations show, without value prior knowledge, FAC's sample efficiency decreases, suggesting that a better value prior can further boost performance. We now explore FAC's robustness to the quality of policy and success reward priors. First, we create noisier policy priors by discretizing each action dimension into $-1, 0, +1$, providing only rough directional information, which we call the discretized policy. We also add uniform noise to the discretized actions at 20% and 50% probabilities. As shown in Fig. 8 (b1), the discretized policy (blue curve) performs similarly to the original (green curve), except for harder tasks like

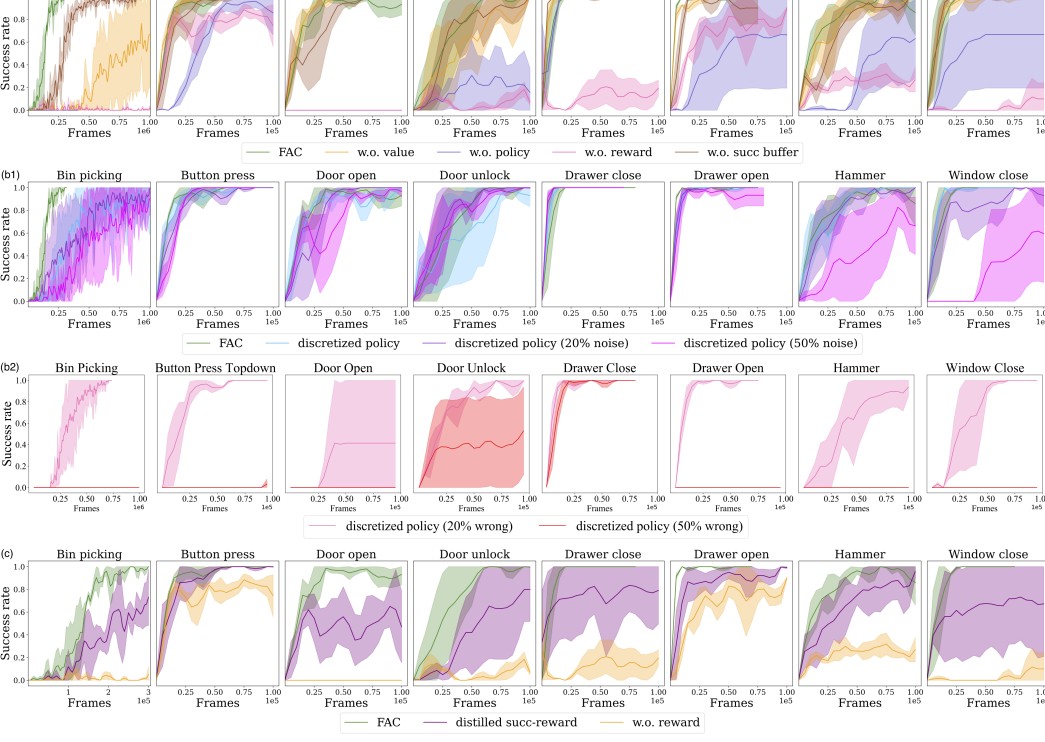

Figure 8: **Results of Ablation Study** (a) Three Foundation Priors and the Success Buffer. (b) The Quality of Policy Prior. (c) The Quality of the success-reward Prior.

bin-picking and door-unlocking, which take more frames to reach 100% success. Adding noise further reduces performance, but even with 50% noise, FAC still achieves 100% success in many environments. We also tested robustness with systematically wrong policy priors, where actions have a 20% or 50% chance of being inverted (e.g., -1 to 1) (Fig. 8 (b2)). FAC performs well with 20% wrong direction but struggles at 50%, where misleading information is abundant. This shows FAC's robustness to moderate systematic noise.

To assess the impact of success-reward prior quality, we distilled a proxy model using 50k offline data from the replay buffer across all 8 tasks. The model, conditioned on task embeddings, has a 1.7% false positive and 9.9% false negative error on the evaluation datasets. We replaced the oracle 0-1 success reward with the predicted reward. As shown in Fig. 8 (c), FAC with the 50k-image proxy model experiences only a limited performance drop compared to the oracle reward and outperforms FAC without success rewards. This demonstrates that FAC performs well even with a noisy success-reward prior. In conclusion, our ablation studies show that FAC is resilient to variations in the quality of foundation priors. The higher the quality, the more sample-efficient FAC becomes.

## 5    Discussion

In this paper, we introduce a novel framework, termed Reinforcement Learning from Foundation Priors (RLFP), which leverages policy, value, and success-reward prior knowledge for RL. Additionally, we also detail the implementation of this concept within actor-critic methods, introducing the Foundation-guided Actor-Critic (FAC) approach. Extensive experiments on real robots and simulated environments demonstrate the effectiveness of RLFP in efficient autonomous learning.

**Limitation and Future Work** This work has limitations, such as relying on human engineering for designing low-level skills, prompts, and code policy demos. Additionally, we fine-tune diffusion models with in-domain data for better prior knowledge. However, as foundation models improve, future methods may only require simple task instructions without fine-tuning. Future exploration can focus on building more accurate and broadly applicable foundation priors and incorporating richer priors into the RLFP framework. For example, humans can predict future states, and integrating the predictive knowledge from dynamic foundation models could enhance policy learning.

**Acknowledgments**

This work is supported by the Ministry of Science and Technology of the People´s Republic of China, the 2030 Innovation Megaprojects "Program on New Generation Artificial Intelligence" (Grant No. 2021AAA0150000). This work is also supported by the National Key R&D Program of China (2022ZD0161700).

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

# A  Appendix

## Appendix Table of Contents

### A.1  Reward Shaping in FAC

In this work, we apply the value prior knowledge in Actor-Critic algorithms in the format of reward shaping. Reward shaping guides the RL process of an agent by supplying additional rewards for the MDP [67, 68, 69]. In practice, it is considered a promising method to speed up the learning process for complex problems. Ng et al. [62] introduce a formal framework for designing shaping rewards. Specifically, we define the MDP $\mathcal{G} = (\mathcal{S}, \mathcal{A}, \mathcal{P}, \mathcal{R})$, where $\mathcal{A}$ denotes the action space, and $\mathcal{P} = \Pr\{s_{t+1}|s_t, a_t\}$ denotes the transition probabilities. Rather than handling the MDP $\mathcal{G}$, the agent learns policies on some transformed MDP $\mathcal{G}' = (\mathcal{S}, \mathcal{A}, \mathcal{P}, \mathcal{R}')$, $\mathcal{R}' = \mathcal{R} + F$, where $F$ is the shaping reward function. When there exists a state-only function $\Phi : \mathcal{S} \rightarrow \mathbb{R}^1$ such that $F(s, a, s') = \gamma\Phi(s') - \Phi(s)$ ($\gamma$ is the discounting factor), the $F$ is called a **potential-based shaping function**. Ng et al. [62] prove that the potential-based shaping function $F$ has optimal policy consistency under some conditions as follows:

**Theorem 1** *[62] Suppose that F takes the form of $F(s, a, s') = \gamma\Phi(s') - \Phi(s)$, $\Phi(s_0) = 0$ if $\gamma = 1$, then for $\forall s \in \mathcal{S}, a \in \mathcal{A}$, the potential-based F preserve optimal policies and we have:*

$$Q^*_{\mathcal{G}'}(s, a) = Q^*_{\mathcal{G}}(s, a) - \Phi(s)$$
$$V^*_{\mathcal{G}'}(s) = V^*_{\mathcal{G}}(s) - \Phi(s) \tag{2}$$

The theorem indicates that the potential-based shaping rewards treat every policy equally. Thus it does not prefer $\pi^*_{\mathcal{G}}$. Moreover, under the guidance of shaping rewards for the agents, a significant reduction in learning time can be achieved. In practical settings, the real-valued function $\Phi$ can be determined based on domain knowledge.

### A.2  Experimental Details of FAC

Since FAC is built upon DrQ-v2, the hyper-parameters of training the actor-critic model are the same as DrQ-v2 [11]. The n-step TD target value and the action in Eq. 1 are as follows, where $\bar{\theta}_k$ are the moving weights for Q target networks.

$$y = \sum_{i=0}^{n-1} \gamma^i r_{t+i} + \gamma^n \min_{k=1,2} Q_{\bar{\theta}_k}(s_{t+n}, a_{t+n}),$$
$$a_t = \pi_\phi(s_t) + \epsilon, \epsilon \sim \text{clip}(\mathcal{N}(0, \sigma^2), -c, c) \tag{3}$$

**Hyper-parameters in FAC** Morever, the observation shape is $84 \times 84$ on real robots and in simulation. For better representations of the scene, we use a wrist camera on real robots. In simulation, we stack 3 frames and repeat actions for 2 steps, while we do not stack frames or repeat actions on real robots. In Meta-World, we follow the experimental setup of [60]. Specifically, the horizon length is set to 125 frames for all tasks except for bin-picking and button-press-topdown, which are set to 175. The total frames of the 8 tasks are 100k, except for the task bin-picking, which is set to 1M. Notably, we set the same camera view of all the tasks for consistency. On real robots, we set the

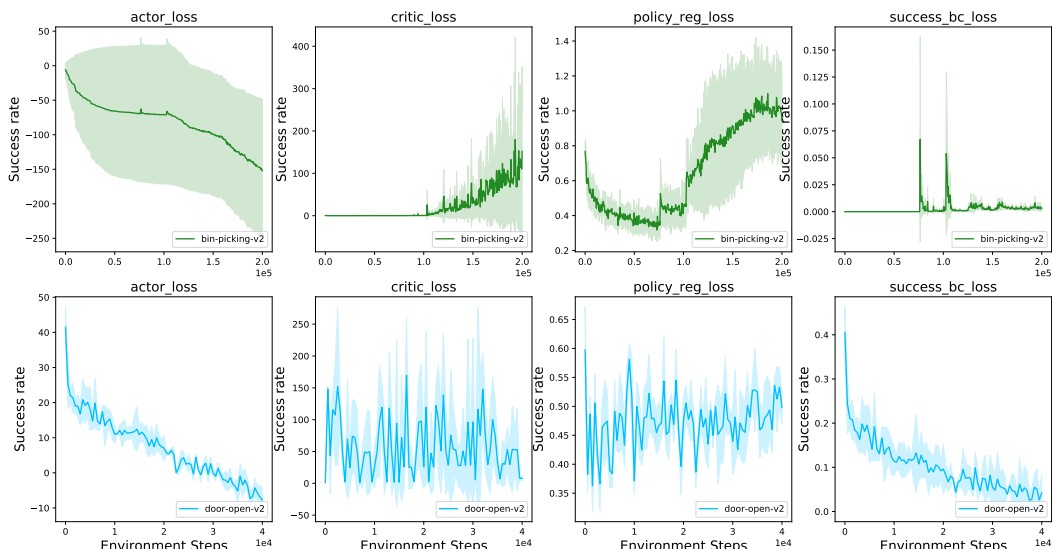

Figure 9: Training loss curves of bin-picking and door-open.

horizon of the 5 tasks (Pick Place, Open Door, Water Plants, Unscrew Bottle Cap, Play Golf) to 40, 40, 50, 60, and 25, respectively. We evaluate them every 10 minutes, and the positions of the objects vary in 5cm in Pick Place, Open Door, Watering Plants, and 3cm in Unscrew Bottle Cap and Play Golf. We reset the objects manually because the current foundation model cannot deal with reset problems. The difference in the hyper-parameters between the real robots and the simulation is as follows:

Table 2: The Main Difference of the Hyper-parameters of FAC on real robots and in simulation.

| Parameter | On real robots | In simulation |
|---|---|---|
| Frame Stack | 1 | 3 |
| Action Repeat | 1 | 2 |
| Seed Frames | 10 Trajectory | 4000 |
| Std of Actor | 0.1 | $1 \to 0.1$ |
| Feature_dim in DrQ-v2 | 512 | 50 |
| UTD ratio | 20 | 1 (N/A) |
| Choose better action under $Q$ | Yes | No |
| Use wrist camera | Yes | No |

The training loss curves of the task bin-picking and door-open for reference, as shown in Fig. 9.

**More Experimental Setup Details on real robots**

Here we give the task description of the 5 tasks on Franka Arm. For the task Unscrew Bottle Cap, we will fix the bottle when the arm tries to unscrew the cap.

- Pick Place: *Pick up the purple eggplant and place it onto the blue plate.*

- Open Door: *There is a white cabinet on the table and there is a cucumber in it. The door can only be opened from the outside. Please open the door by a large margin so that the cucumber can be seen.*

- Watering Plants: *There are a blue watering kettle and a potted plant. Please help me watering the plant and keep watering still.*

- Unscrew Bottle Cap: *There are a plastic bottle with a green cap and a pink plate on the table. The bottle will be fixed on the table, so that you cannot lift the bottle. Please help me*

*unscrew the bottle cap and place it on the pink plate. You mush know how to unscrew the bottle cap, anticlockwise or clockwise?*

- Play Golf: *There is a golf hole and a golf ball on the table. The robotic arm is tied by a golf club. Please shoot the golf ball into the set.*

The ground-truth success reward functions of each task are as follows, which is leveraged in evaluation.

- Pick Place: *dist(xy position of the eggplant, xy position of the blue plate center) < radius & dist(z of the eggplant, z of the blue plate) < ϵ, ϵ = 0.01.*
- Open Door: *dist(xy position of the handle, xy position of the cabinet center) > ϵ, ϵ = 0.1.*
- Watering Plants: *dist(z of the sprout, z of the plants) > $\epsilon_1$ & dist(xy of the sprout, xy of the plants) < radius & abs(horizontal orientation of the sprout) < $\epsilon_2$ degree, $\epsilon_1 = 0.1, \epsilon_2 = 15$.*
- Unscrew Bottle Cap: *dist(xyz of the cap, xyz of the bottle center) > ϵ & (cap in the plate), ϵ = 0.15. ('cap in the plate' is the same as the function in pick place).*
- Play Golf: *(golf in the golf hole). ('golf in the golf hole' is the same as the function in pick place).*

**Reward functions for vanilla RL** Since it is hard to get the object state in real time, we assume the object state will be attached to the robot state after some actions. For example, we have the initial position of the eggplant, the position, and the size of the target plate. After the gripper reaches the pos of the eggplant and takes action "grasp", the pos of the eggplant will be set the same as the eef pos. The reward function is the normal pick-place reward function. For the door-open, we get the handle initial position, and once the arm is reaching in the bound of 5cm of the handle and closes the gripper, the position of the handle will be attached to the eef of the robot. Therefore, the reward functions are based on the distance between the gripper and some target position sequences, including the position distance and the orientation distance. We give the reward functions of the Door Open and Play Golf as examples, which are harder tasks.

Open Door: $R_{\text{approach}} = \max\left(0, 1 - tanh(10 \times \mathbf{p}_{\text{gripper}} - \mathbf{p}_{\text{handle}}\|)\right)$, encourages the robot to get closer to the handle by diminishing rewards as the distance decreases. $R_{\text{orient}} = 1 - \frac{\|\mathbf{q}_{\text{gripper}} - \mathbf{q}_{\text{handle}}\|}{2}$, incentivizes aligning the robot's gripper orientation with that of the handle. $R_{\text{pull}} = \max\left(0, \theta_{\text{door}} - \theta_{\text{prev\_door}}\right)$, increases with the door's opening angle. $R_{\text{open}} = \min\left(\frac{\theta_{\text{door}}}{90.0}, 1.0\right)$, prioritizes achieving a 90-degree opening. A small time penalty, $R_{\text{time}} = -0.01$, is applied per time step to encourage faster task completion. The total reward is then formulated as: $R_{\text{total}} = 0.01 * R_{\text{approach}} + 0.05 * R_{\text{orient}} + 0.1 * R_{\text{pull}} + 0.1 * R_{\text{open}} + R_{\text{time}}$.

Play Golf: We use the golf club's length and the end-effector's (eef) position to determine the clubhead's state. The ball back position is defined as the back of the ball, which is the optimal point to strike. $R_{\text{approach}} = \max\left(0, 1 - \tanh(10 \times \|\mathbf{p}_{\text{clubhead}} - \mathbf{p}_{\text{ball\_back}}\|)\right)$, encourages the robot to position the clubhead at the back of the ball. $R_{\text{orient}} = 1 - \frac{\|\mathbf{q}_{\text{clubhead}} - \mathbf{q}_{\text{target}}\|}{2}$, which incentivizes proper alignment of the clubhead for an accurate strike. Additionally, the reward for the clubhead's velocity at the moment of impact, $R_{\text{velocity}} = \max\left(0, 1 - \tanh(10 \times \|\mathbf{v}_{\text{clubhead}} - \mathbf{v}_{\text{desired}}\|)\right)$, ensures that the clubhead reaches the optimal speed to strike the ball effectively. So the final reward is: $R_{\text{total}} = 0.1 * R_{\text{approach}} + 0.5 * R_{\text{orient}} + 0.5 * R_{\text{velocity}} + R_{\text{time}}$.

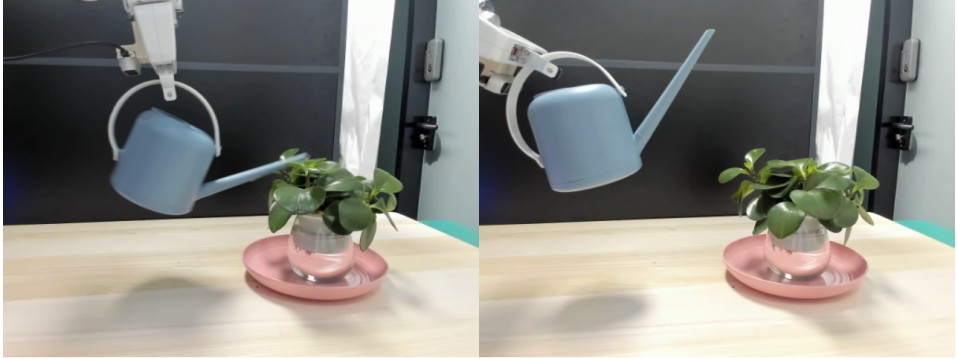

1 The robotic arm in the image appears to have positioned the spout of the watering can directly over the plant, which suggests that it is in the process of watering the plant. This setup is typically indicative of an automated or robotic system designed to water plants without human intervention.

0 The spout of the watering can is not oriented horizontally over the plant; rather, it is tilted at an angle that suggests it may not be actively watering the plant at this moment. It appears to be in a position that might be before or after the watering action, or possibly in a paused state.

Figure 10: Example of success identification by GPT-4V in task Watering Plants. Given the question. *Does the robotic arm water the plants? Attention, if the spout orients horizontally over the plant, you should output 1 for yes. Otherwise, you should output 0 for no without any space first. Be sure of your answer and explain your reason afterward.* The foundation model can give the correct success reward as well as the corresponding explanations.

## A.3   Details of Acquring Foundation Priors

### A.3.1   Acquring Foundation Priors on Franka

**Examples of Success-reward Discrimination Prompts** It is significant to feed the success-reward model by high-resolution images, which we choose $512 \times 512$. Here we give an example of the success-reward prompts in FAC towards the task Watering Plants, by leveraging GPT-4V, as follows: *Does the robotic arm water the plants? Attention, if the spout orients over horizontally over the plant, you should output 1 for yes. Otherwise, you should output 0 for no without any space first. Be sure of your answer and explain your reason afterward.* Then we can receive the corresponding success-reward as well as the explanations, shown in Fig. 10.

**Examples of Code Policy Prompts.** Before code generation, we define some primitive skills. We implement the corresponding interface between primitive skills and control systems, so that they can be executed directly by the robot. The primitive low-level skills and the corresponding params are as follows:

- move_to x y z: move the gripper to the position (x, y, z).
- grasp: grasp with the gripper.
- release: release the gripper.
- rotate_clockwise: rotate gripper clockwise by 90 degree.
- rotate_anticlockwise: rotate gripper anticlockwise by 90 degree.
- orient_half_horizontally: change the orientation of the gripper half horizontally to make it grasp more easily.
- orient_vertically: change the orientation of the gripper vertically to make ti grasp something vertically.
- strike_front: strike something to the front of the arm.
- strick_back: strike something to the back of the arm.

Firstly, we input prompts that include a task description, the initial scene, and the expected format for the generated code. Notably, the task used in the example differs from the five tasks we evaluate. For

the downstream tasks, we rely on the GPT-4V to generate the code policy based on new images and the corresponding task descriptions, shown in Fig. 11. We also specify the range of object positions within the scene, enabling the VLM to estimate the position of the target object and generate code that includes skill plans and position parameters. Limited to the current ability of GPT-4V, it can build reasonable code policies under small changes in object positions.

Then, we feed in the current image and the task descriptions: *There are a plastic bottle with a green cap and a pink plate on the table. The bottle will be fixed on the table, so that you cannot lift the bottle. Please help me unscrew the bottle cap and place it on the pink plate. You mush know how to unscrew the bottle cap, anticlockwise or clockwise?* Then we can receive the **generated code policy** as shown in the code text, which can make the correct rotation to unscrew the cap.

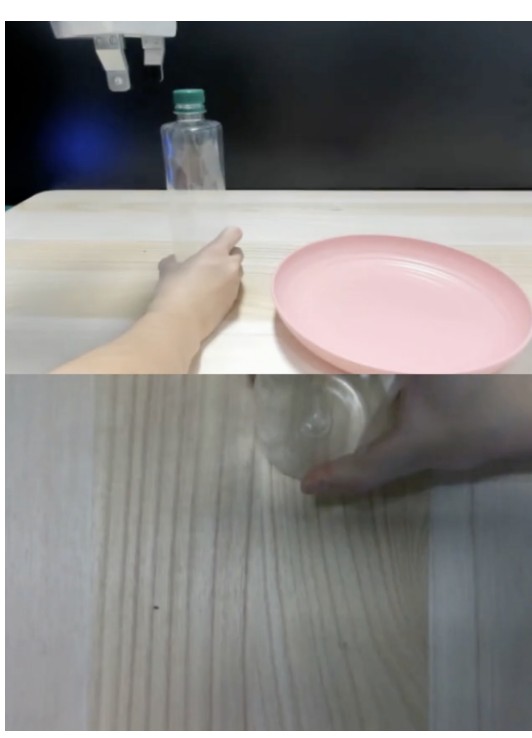

Figure 11: Initial observation image of the task Unscrew Bottle Cap, including the fixed camera view and the wrist camera view.

On real robots, the action space is 7-dim, consisting of the delta position (x, y, z) of the gripper, the delta orientation (roll yaw pitch) of the gripper, and the grasp/release of the gripper. We use the OSC control mode. As for the policy output of the code policy, it generates plans like "move_to x1 y1 z1". Then we use robotics control toolkits to compute the delta position/orientation for the robot, which is the prior action. Once the plan "move_to x1 y1 z1" is finished (close to epsilon in practice), the code policy will execute the next plan. In this way, we can acquire the corresponding prior action in each step. The KL constraint is between the prior action and the RL policy. Consequently, the GPT-4V generates code plans that consist of the primitive skills, and then the primitive skill can generate delta prior actions by robotics control codes.

**Policy Description Prompts for Unscrew Bottle Cap**

```
# Query: pick up the plastic bottle
# Some info... (skip here)
# Here are the skills that you can choose:
# 1. move_to x y z: move the gripper to
    the position of (x, y, z)
# 2. grasp: grasp with the gripper
# 3. release: release the gripper
# 4. rotate_clockwise: rotate the gripper
    clockwise by only 90 degree
# 5. rotate_anticlockwise: rotate the
    gripper anticlockwise by 90 degree
# 6. reset: move back to the initial
    position
# Attention: For safety, you cannot rotate
     clockwise or rotate
# anticlockwise twice continuously. Think
    carefully about how to
# finish the task.

def code_policy():
    # Obtain the position of plastic
        bottle. Estimate the position.
    object_pos = [0.5, 0., 0.12]
    x1, y1, z1 = object_pos
    # Pick up
    plans = [
        # move above the bottle
        f'move_to {x1:.2f} {y1:.2f} {z1 +
            0.05:.2f}',
        # move to the bottle
        f'move_to {x1:.2f} {y1:.2f} {z1:.2
            f}',
        # grasp the bottle
        'grasp',
        # lift the cucumber
        f'move_to {x1:.2f} {y1:.2f} {z1 +
            0.2:.2f}',
    ]
    return plans
```

**Generated Code Policy for Unscrew Bottle Cap**

```
def code_policy():
    # Bottle cap position (estimated near
        the top of the bottle: height of
        the bottle + table z-axis)
    cap_pos = [0.5, 0., 0.26]
    x1, y1, z1 = cap_pos

    # Plate position (estimated to be at
        the provided position)
    plate_pos = [0.75, 0., 0.06]
    x2, y2, z2 = plate_pos

    # Plans to unscrew the bottle cap and
        place it on the plate
    plans = [
        # Move above the bottle cap
        f'move_to {x1:.2f} {y1:.2f} {z1 +
            0.05:.2f}',
        # Move to the bottle cap
        f'move_to {x1:.2f} {y1:.2f} {z1:.2
            f}',
        # Grasp the bottle cap
        f'grasp',
        # Rotate anticlockwise to unscrew
        f'rotate_anticlockwise',
        # Lift the cap
        f'move_to {x1:.2f} {y1:.2f} {z1 +
            0.05:.2f}',
        # Move above the plate
        f'move_to {x2:.2f} {y2:.2f} {z1 +
            0.05:.2f}',
        # Move down to the plate
        f'move_to {x2:.2f} {y2:.2f} {z2:.2
            f}',
        # Release the cap onto the plate
        f'release',
        # Move away
        f'move_to {x2:.2f} {y2:.2f} {z2 +
            0.2:.2f}'
    ]
    return plans
# done
```

### A.3.2 Acuqring Foundation Priors on Meta-world

**Training Inverse Dynamics Model** We build the inverse dynamics model $\rho(s_t, s_{t+1})$ as follows:

- Takes inputs as $s_t, s_{t+1}$, with the shape of $3 \times 84 \times 84$.
- A Downsample Model, which outputs the representation with the shape of $128 \times 2 \times 2$.
- Flatten the planes into 512-dimension vectors.
- 1 Linear layer with ReLU, which outputs the 64-dimension vectors.
- 1 Linear layer with ReLU, which outputs the 64-dimension vectors.
- 1 Linear layer with ReLU, which outputs the action dimension vectors (equal to 4).

The Downsample model is designed as follows:

- 1 convolution with stride 2 and 128 output planes, output resolution $42 \times 42$. (ReLU)
- 2 residual block with 128 planes.
- Average pooling with stride 2 (kernel size is 3), output resolution $21 \times 21$. (ReLU)
- 2 residual block with 128 planes.
- Average pooling with stride 3 (kernel size is 5), output resolution $7 \times 7$. (ReLU)
- 2 residual block with 128 planes.
- Average pooling with stride 3 (kernel size is 4, no padding), output resolution $2 \times 2$. (ReLU)

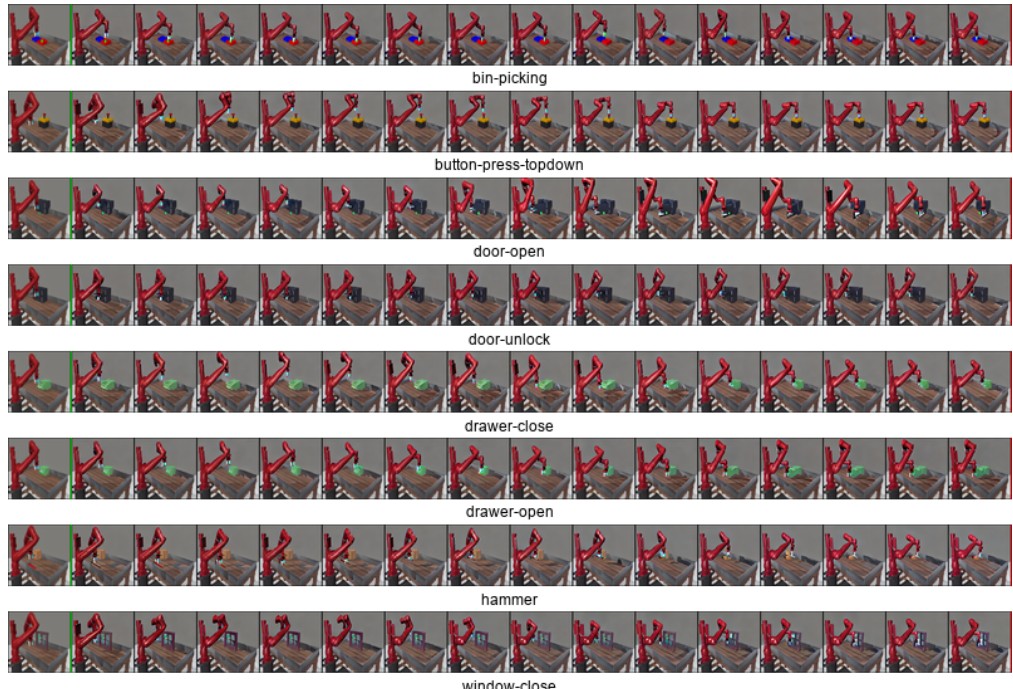

Figure 12: The generated videos from the diffusion model Seer given the initial images as well as task descriptions.

We use the 1M replay buffer trained from vanilla DrQ-v2 for each task and collect them together as the dataset.

**Distilling Diffusion Policy Foundation Models** We use the fine-tuned VLM Seer to collect 100 videos for each task (1000 in bin-picking-v2), and use the trained inverse dynamics model $\rho(s_t, s_{t+1})$ to label pseudo actions for the videos. The example of generated videos is illustrated in Fig. 12. Then, we do supervised learning to train the policy foundation prior model under the dataset, which is conditioned on the task. For convenience, we encode the task embedding as a one-hot vector, which labels the corresponding task. Thus, the size of the task embedding is 8. Here, the architecture of the distilled policy model is as follows, where the downsample model is the same as that in the inverse dynamics model.

- Takes inputs as $s_t, e_t$, with the shape of $3 \times 84 \times 84$ and $1 \times 8$.

- A Downsample Model, which outputs the representation with the shape of $128 \times 2 \times 2$.

- Flatten the planes into 512-dimension vectors.

- Concat the 512 vector and the task embedding into 520-dimension vectors.

- 1 Linear layer with ReLU, which outputs the 64-dimension vectors.

- 1 Linear layer with ReLU, which outputs the 64-dimension vectors.

- 1 Linear layer with ReLU, which outputs the action dimension vectors (equal to 4).

The training hyper-parameters of the inverse dynamics model $\rho(s_t, s_{t+1})$ and the distilled policy model $M_\pi(s_t, \mathcal{T})$ are in Table 3. The hyper-parameters of training FAC agents are the same as DrQ-v2 [11].

Table 3: Hyper-parameters for Building the Policy Foundation Models in Meta-World.

| Parameter | Training $\rho$ | Training $M_\pi$ |
|---|---|---|
| Minibatch size | 256 | 256 |
| Optimizer | AdamW | AdamW |
| Optimizer: learning rate | 1e-4 | 5e-4 |
| Optimizer: weight decay | 1e-4 | 1e-4 |
| Learning rate schedule | Cosine | Cosine |
| Max gradient norm | 1 | 1 |
| Training Epochs | 50 | 300 |

## A.4 Running Clock Time Analysis

### A.4.1 Analysis of Extra Running Clock Time in FAC

Since it brings great performance and efficiency increases for RL by leveraging the foundation prior knowledge, it is interesting and significant to investigate the clock time of running the foundation models in practice. Here we discuss the time consumed by each part from the foundation models in real and simulation. In practice, some foundation models are accessed only at the beginning or at the final step, which results in a much smaller amortization time over the trajectory. For example, GPT-4V detects success only in the final step. To make fair comparisons, we record the running clock time per trajectory and get the amortized time per step. We set the trajectory length to 50 steps for convenience. To make clear comparisons, we conclude the operations without access to foundation models into *Normal Operation* part, and those operations with access to foundation models into *Foundation Model Operation* part. All the results are evaluated in a single 3090 GPU, and we calculate the average clock time during training among 50 trajectories on average from all the tasks.

Tab. 4 and 5 are the results of real robots and simulations concerning clock time during training. Here, **Action Move** means the process of taking actions by the robotic arm. The **Action Inference** means the inference time of the learned policy model. Notably, in the simulation, we offline distill a prior policy from the diffusion model due to the heavy time-cost of video generation (in Sec. 3.3). Thus, the clock time of the Diffusion Policy Prior for video generation is not included in the training time, although it is much more time-consuming than the other operations in simulation.

Generally, in both environments, the time complexity of running foundation models is about 2 times more than the vanilla setting in amortization. And the most time-consuming part is the **Action**

Table 4: **Running Time of Each Part in FAC on Real Robots**. On real robots, the most time-consuming part of the foundation model operation is generating policy code from GPT-4V. The total amortized time of foundation model operations is a little larger to the normal operations.

| Clock Time on **Real Robots** (s) | Per Trajectory | Amortized Step |
|---|---|---|
| *Normal Operation* | | |
| **Action Move** of the gripper in the real world | - | **0.320** |
| **Action Inference** from policy $\pi(a|s)$ | - | 0.001 |
| **Total** | - | 0.321 |
| *Foundation Model Operation* | | |
| **Code Policy Prior**: policy code generation from GPT-4V | 14.76 | 0.290 |
| **Code Policy Prior**: action generation by code policy | - | 0.001 |
| **Success Prior**: from GPT-4V | 4.48 | 0.089 |
| **Value Prior**: value inference from VIP | - | 0.013 |
| **Total** | - | 0.393 |

Table 5: **Running Time of Each Part in FAC on Meta-World**. In the simulation, the most time-consuming part of the foundation model operation is the value inference from VIP. The total amortized time of foundation model operations is smaller than the normal operations.

| Clock Time in **Simulation** (s) | Per Trajectory | Amontized Step |
|---|---|---|
| *Normal Operation* | | |
| **Action Move** of the gripper in simulation | - | **0.009** |
| **Action Inference** from policy $\pi(a\|s)$ | - | 0.001 |
| **Total** | - | 0.010 |
| *Foundation Model Operation* | | |
| **Diffusion Policy Prior**: video generation (**done offline**) | ~~11.62~~ | ~~0.230~~ |
| **Diffusion Policy Prior**: action generation | - | 0.001 |
| **Success Prior**: from the success model | - | 0.001 |
| **Value Prior**: value inference from VIP | - | 0.007 |
| **Total** | - | 0.008 |

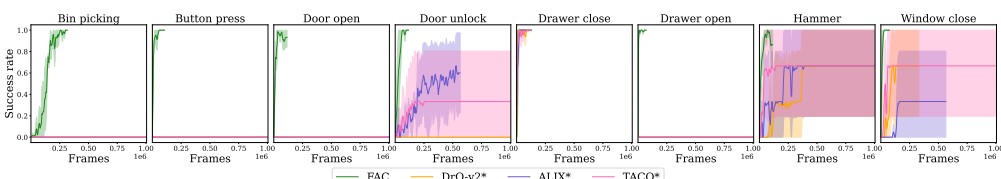

Figure 13: Here '*' in DrQ-v2, ALIX, and TACO means only the 0-1 success reward is provided from the environment, which is different from the original settings in their works. FAC can work for all the tasks while the other baselines fail in half of them. It is significant and sample-efficient to utilize prior knowledge for reinforcement learning.

**Move**. In foundation model operations, generating the code policy from GPT-4V is the slowest on real robots, while the value inference is the slowest in simulation.

Moreover, we record the total training time spent by FAC and vanilla DrQ-v2 to make comparisons. We average the 3 seeds running for each task where we train about 100k steps on a single 3090 GPU. Vanilla DrQ-v2 takes 32m 2s on average; while FAC takes 1h 11min 17s on average, which is 2 times more than that without foundation priors. As shown in Fig. 6, even at the same clock time, FAC is better than vanilla DrQ-v2 with ground-truth rewards. FAC can solve 7/8 tasks in 100k frames, while the vanilla DrQ-v2 can only solve 2/8 in 200k frames. Furthermore, FAC can solve more tasks than the baselines. Consequently, we conclude that our method significantly outperforms the baselines concerning training clock time.

### A.5  More Ablations Results

**Comparison to More Baselines with Success-reward Only** Here, we also add some baselines under the setting, where only the success-reward foundation prior is provided. We choose the recent SOTA model-free RL algorithms on Meta-World ALIX [70] and TACO [71], as well as the baseline DrQ-v2 [11] with the success-reward only. Notably, ALIX and TACO are both built on DrQ-v2. The results are shown in Fig. 13, where '*' means that only a 0-1 success reward is given. Only FAC can achieve 100% success rates in all the environments. DrQ-v2*, ALIX*, and TACO* can not work on hard tasks such as bin-picking and door-open. FAC requires fewer environmental steps to reach 100% success rates, as shown in the Figure. The results on the new baselines can verify the significance and efficiency of utilizing the abundant prior knowledge for RL in a way.

**Comprasion to BC Policies from Policy Prior** Another interesting baseline of leveraging foundation policy prior is to collect success demonstrations by the prior policy $M_\pi$ and train a behavior cloning policy. Here we collect 100 successful demonstrations on Meta-World for each task by the

Table 6: The performance comparison between the distilled policy prior and the learned bc policy on Meta-World.

|  | Prior Policy $M_\pi$ | BC Policy | FAC |
|---|---|---|---|
| bin-picking-v2 | 0 | 0 | 1.00 |
| button-press-topdown-v2 | 0.45 | 0.15 | 1.00 |
| door-open-v2 | 0 | 0 | 1.00 |
| door-unlock-v2 | 0 | 0 | 1.00 |
| drawer-close-v2 | 0.10 | 0.10 | 1.00 |
| drawer-open-v2 | 0.05 | 0 | 1.00 |
| hammer-v2 | 0.15 | 0.10 | 1.00 |
| window-close-v2 | 0.30 | 0.05 | 1.00 |

prior policy $M_\pi$. Noticing that the policy prior achieves 0% success rate on some tasks, we collect 100 trajectories then. The results are in Tab. 6. Generally, we find that the learned BC policy cannot outperform the prior policy itself in all the tasks. More significantly, in some tasks, such as windowclosev2, the BC policy achieves much worse results than the prior policy. This is because the prior policy works only in some certain scenarios, which introduces bias in the collected demos. Although the prior policy fails in some scenarios, it can introduce some informative actions, which can be much better than random actions. Therefore, it is more reasonable to leverage the prior actions as guidance for RL.

## A.6 Proof of the Optimality under Policy Regularization

**Lemma 1** *The policy $\pi_m = \frac{1}{1+\beta}\hat{\pi}_{\phi_m} + \frac{\beta}{1+\beta}M_\pi$, is the solution to the optimization problem of the actor shown in Equation 1.*

**Proof 1** *First, $\hat{\pi}_{\phi_m}$ is the RL policy optimized by standard RL optimization problem in m-th iteration, illustrated in the following equation.*

$$\hat{\pi}_{\phi_m} = \arg\max_{\hat{\pi}_\phi} \mathbb{E}_{\tau \sim \hat{\pi}_\phi}[Q(s,a)] \quad as\ m \to \infty \tag{4}$$

*Note that the following derivation omits the variance of Gaussian distribution for convenience. This is because the variance is independent of the state in the deterministic Actor-Critic algorithms DrQ-v2 algorithm.*

*According to Equation 1, the policy $\pi_m$ can be represented as:*

$$\pi_m = \arg\min_{\pi}[-\mathbb{E}_{\tau \sim \pi}Q(s,a) + \beta\boldsymbol{KL}(\pi, M_\pi)] \tag{5}$$

*Adding $\mathbb{E}_{\tau \sim \hat{\pi}_{\phi_m}}Q(s,a)$ in Equation 5, we can rewrite it as:*

$$\pi_m = \arg\min_{\pi}[\mathbb{E}_{\tau \sim \hat{\pi}_{\phi_m}}Q(s,a) - \mathbb{E}_{\tau \sim \pi}Q(s,a) + \beta\boldsymbol{KL}(\pi, M_\pi)] \tag{6}$$

*Considering $\mathbb{E}_{\tau \sim \hat{\pi}_{\phi_m}}Q(s,a)$ is not related to the optimization objective, the above equation holds. Intuitively, we can observe that there exist two parts in the objective. About the first part, we can use importance sampling to obtain:*

$$\mathbb{E}_{\tau \sim \hat{\pi}_{\phi_m}}Q(s,a) - \mathbb{E}_{\tau \sim \pi}Q(s,a) = \mathbb{E}_{\tau \sim \hat{\pi}_{\phi_m}}[\frac{\hat{\pi}_{\phi_m} - \pi}{\hat{\pi}_{\phi_m}}Q(s,a)] \tag{7}$$

*Since $\hat{\pi}_{\phi_m}$ can be represented as $\arg\max_{\hat{\pi}_\phi} \mathbb{E}_{\tau \sim \hat{\pi}_\phi}[Q(s,a)]$ when $m$ approaching to infinity, the minimum of $\mathbb{E}_{\tau \sim \hat{\pi}_{\phi_m}}Q(s,a) - \mathbb{E}_{\tau \sim \pi}Q(s,a)$ can be achieved when the minimum of the following equation exits.*

$$\arg\min_{\pi}\|\hat{\pi}_{\phi_m} - \pi\| \iff \arg\min_{\pi}\|\arg\max_{\hat{\pi}_\phi} \mathbb{E}_{\tau \sim \hat{\pi}_\phi}[Q(s,a)] - \pi\| as\ m \to \infty \tag{8}$$

*Let us see the second part in Equation 6. $\pi$ and $M_\pi$ are Gaussian distributions and the variances of distributions are constant in our framework. Thus, $\boldsymbol{KL}(\pi, M_\pi) \iff \|\pi - M_\pi\|$ holds.*

*Hereafter, we can reformulate Equation 6 as follows:*

$$\pi_m = \arg\min_{\pi}[\|\arg\max_{\hat{\pi}_\phi}\mathbb{E}_{\tau\sim\hat{\pi}_\phi}[Q(s,a)] - \pi\| + \beta\|\pi - M_\pi\|] \tag{9}$$

*Based on the Lemma 1 in [72], the solution to the above problem is derived as:*

$$\pi_m = \frac{1}{1+\beta}\hat{\pi}_{\phi_m} + \frac{\beta}{1+\beta}M_\pi \tag{10}$$

*To this end, the policy $\pi_m$ is the solution to the proposed optimization problem in this paper.*

**Theorem 2** *Let $D_{sub} = D_{TV}(\pi_{opt}, M_\pi)$ be the bias between the optimal policy and the prior policy, the policy bias $D_{TV}(\pi_m, \pi_{opt})$ in m-th iteration can be bounded as follows:*

$$D_{TV}(\pi_m, \pi_{opt}) \geq D_{sub} - \frac{1}{1+\beta}D_{TV}(\hat{\pi}_{\phi_m}, M_\pi)$$
$$D_{TV}(\pi_m, \pi_{opt}) \leq \frac{\beta}{1+\beta}D_{sub} \quad as\ m \to \infty \tag{11}$$

**Proof 2** *Note that the following derivation is most inspired by Theorem 1 in [72]. According to Lemma 1, the policy $\pi_m$ can be represented as $\frac{1}{1+\beta}\hat{\pi}_{\phi_m} + \frac{\beta}{1+\beta}M_\pi$.*

*Then, let us define the policy bias as $D_{TV}(\pi_m, \pi_{opt})$, and $D_{sub} = D_{TV}(\pi_{opt}, M_\pi)$. Since $D_{TV}$ is a metric that represents the total variational distance, we can use the triangle inequality to obtain:*

$$D_{TV}(\pi_m, \pi_{opt}) \geq D_{TV}(M_\pi, \pi_{opt}) - D_{TV}(M_\pi, \pi_m) \tag{12}$$

*According to the mixed policy definition in Equation 10, we can further decompose the term $D_{TV}(M_\pi, \pi_m)$:*

$$\begin{aligned}
D_{TV}(M_\pi, \pi_m) &= \sup_{(s,a)\in S\times A}\left|M_\pi - \frac{1}{1+\beta}\hat{\pi}_{\phi_m} - \frac{\beta}{1+\beta}M_\pi\right| \\
&= \frac{1}{1+\beta}\sup_{(s,a)\in S\times A}|\hat{\pi}_{\phi_m} - M_\pi| \\
&= \frac{1}{1+\beta}D_{TV}(\hat{\pi}_{\phi_m}, M_\pi)
\end{aligned} \tag{13}$$

*This holds for all $m \in \mathbb{N}$ from Equation 12 and Equation 13, and we can obtain the lower bound as follows:*

$$D_{TV}(\pi_m, \pi_{opt}) \geq D_{sub} - \frac{1}{1+\beta}D_{TV}(\hat{\pi}_{\phi_m}, M_\pi) \tag{14}$$

*The RL policy $\hat{\pi}_{\phi_m}$ can achieve asymptotic convergence to the (locally) optimal policy $\pi_{opt}$ through the policy gradient algorithm. In this case, we can derive the bias between the mixed policy $\pi_m$ and the optimal policy $\pi_{opt}$ as follows:*

$$\begin{aligned}
D_{TV}(\pi_{opt}, \pi_m) &= \sup_{(s,a)\in S\times A}\left|\pi_{opt} - \frac{1}{1+\beta}\hat{\pi}_{\phi_m} - \frac{\beta}{1+\beta}M_\pi\right| \\
&= \frac{\beta}{1+\beta}\sup_{(s,a)\in S\times A}|\pi_{opt} - M_\pi| \quad as\ m \to \infty \\
&= \frac{\beta}{1+\beta}D_{TV}(\pi_{opt}, M_\pi) \quad as\ m \to \infty \\
&= \frac{\beta}{1+\beta}D_{sub} \quad as\ m \to \infty
\end{aligned} \tag{15}$$

*Therefore, we obtain the upper bound.*

