# OpenReview forum: "Reinforcement Learning with Foundation Priors: Let Embodied Agent Efficiently Learn on Its Own"
_robot-learning.org/CoRL/2024/Conference — CoRL 2024_

### Official Review · Reviewer_bb5a · 2024-07-18

**Originality:** 2
**Technical Quality:** 3
**Clarity Of Presentation:** 3
**Potential Impact:** 2
**Recommendation:** 3
**Confidence:** 4

**Review:**

The foundation models used to ease reinforcement learning comprise a policy prior, which uses GPT4 together with a code policy defining a set of useful movements, a value prior derived from VIP trained on general robotic tasks, and a success prior where GPT4 was prompted to judge completion. This approach is evaluated on a range of real-world and simulated tasks.

### Strengths:
- Good results and extensive experiments.

### Weaknesses:
- The proposed methods are too simple to be called a framework. This work is just reusing and combining prior work.
- Experiments needs an additional ablation. (see below)
- requires code policy.
    - This makes this algorithm hard to extend to more complex tasks.

### Suggestions:
- Just present it as the FAC algorithm, rather than a framework. There is no theory nor novelty behind it to be a useful framework.
- Table 1:
    - add DrQ-v2 results (even if they are 0) ; also what rewards have been used there?
    - I think another experiment is require in which DrQ is initialized not from a random policy, but one which is pretrained using BC (it is common to warm-start a policy). I think you can use the action generated from code-policy from GPT4.
    - maybe for the vanilla one, one can check whether success rewards  work better that the handcrafted ones if the policy has a warm start.
- Describe what code policy means in more detail in the main paper and make it clear that a detailed description can be found in the appendix. I found it by luck; people usually don’t spend much time reading the appendix!

### Post-Rebuttal
- my concerns were addressed, which is why I updated my score.
- while I still not fully agree that the introduction of the framework is actually needed, I do understand their perspective.
- happy to see the additional ablations were done during the rebuttal.

**Quality Of The Limitations Section:**

3

**Questions For Rebuttal:**

See above.

**Robotics Focus:**

4

**Summary Of Paper:**

This paper proposes an algorithm using three foundation models to ease reinforcement learning for real-world tasks.

**Summary Of Recommendation:**

I think that this paper requires a bit more work regarding the experiments. Also I would change the storyline, just combining a few common techniques is not enough for calling it a general framework. I would just calling it an algorithm. Hence, I am voting for rejection, but I am willing to raise my score if my concerns are addressed.

---

### Official Review · Reviewer_kNo7 · 2024-07-20
**A Comprehensive Evaluation of Policy, Reward, and Value Priors for RL training**

**Originality:** 3
**Technical Quality:** 5
**Clarity Of Presentation:** 5
**Potential Impact:** 4
**Recommendation:** 4
**Confidence:** 4

**Review:**

Recently, many papers have proposed different usages of foundation models for improving RL training, but most focus on a single aspect - reward function, prior policy, etc. This paper tries to incorporate all of them with an ablation study that shows how each component complements the others and leads to even better sample efficiency. The main real-world results are impressive, resulting in the training of a policy from scratch in under an hour.

The main downside of the work is that the priors, especially the policy, still heavily rely on human engineering in designing the low-level skills and the prompts. From the examples in the appendix, the prompts are highly tailored to the task - "There is a plastic bottle with a green cap and a pink plate on the table," or "If the spout orients horizontally over the plant, you should output 1". In addition, writing these prompts usually requires a few iterations until they work well. Therefore, while it still shows impressive sample efficiency, the method doesn't remove the need for human engineering.

**Quality Of The Limitations Section:**

3

**Questions For Rebuttal:**

- In section 4.1, you mention, "the positions of objects vary within a predefined range across different trajectories." Can you provide more information on that? In addition, have you tried to vary the position beyond the training range to check for generalization capabilities? In general, the paper lacks generalization experiments.
- For section 4.1, while the wall clock is indeed the correct value for comparison, please also provide the number of real-world trajectories that were collected during this time. It is important to understand the sample efficiency of the algorithm.
- One thing that is missing is the evaluation of the prior success reward. How accurate is it? Can you provide an analysis, either in an environment where you have access to the GT or using a human observer.
- You mentioned that the GPT-4V prior policy generate a trajectory based only on the initial state. If so, how do you take the KL loss between the policy and prior policy actions on non-initial states? You don't have the prior policy actions for these states.
- The idea of keeping a success buffer and adding a BC loss to it doesn't have anything to do with foundation models and can be applied with any reward. However, it is not a common practice in off-policy RL. Why do you believe that it should be used here? Especially when your ablation shows that it helps in only one of the tasks you consider (Bin Picking).

**Robotics Focus:**

4

**Summary Of Paper:**

The paper explores the best ways to incorporate foundation models into training a new RL policy. It explores three kinds of priors the foundation model can provide - policy, reward, and value. The paper performs a comprehensive study of these components in simulation and the real world. Utilizing all the components leads to an impressive sample efficiency, with the algorithm learning real-world policies in 1 hour of interaction with a new environment.

**Summary Of Recommendation:**

Overall, the paper's contribution and results are solid, and while still limited, they provide an important step in incorporating foundation models into RL training pipelines.

---

### Official Review · Reviewer_M4P9 · 2024-07-21
**The paper brings together interesting ideas and show clear results. However, there are missing details, some of them crucial. Moreover,  limitations are not addressed but written as future work.**

**Originality:** 3
**Technical Quality:** 4
**Clarity Of Presentation:** 3
**Potential Impact:** 3
**Recommendation:** 4
**Confidence:** 5

**Review:**

Strengths:
- The main idea and motivations, on a high level, is described clearly.
- The approach is sound and the combination of the acquired "priors" are novel. (with some caveats on the foundation model assumptions, see below)
- Authors perform both real and simulated evaluations, with strong results and sufficient ablations.

The rest will concentrate on the weaknesses and missing information, mostly in the order of reading:

Title: The title is convoluted. This is also my personal opinion but I do not like the part after the colon character (:). This style is overused. In addition, it is not informative; RL already implies learning "on its own". Maybe the authors wanted to imply "no human demonstrations". Lastly, I am not sure if using prior is the best word. The policy prior is an actual prior. However, the value and success priors are not really priors. The former is the value estimate used for reward shaping and the latter is the main success reward signal, and both of them do not change during learning. However, this is not a strong objection.

Introduction:
- There have been robotic applications of RL going back more than three decades. The first paragraph of introduction sort of implies otherwise.
- Humans also leverage their innate abilities to acquire skills in addition to accumulated knowledge and common sense.
- Even though the human example demonstrates the idea well, it is not really correct. An example with a baby and toys would have been better. There are interesting papers by Meltzoff et al, mainly involving imitation (there are learning aspects which align with the presented idea) that can be used.
- The claim of "agnostic to foundation models" needs some expansion. The method definitely requires well trained models. The authors find that some of the foundation models are not good enough and fine-tune them. Existence of strong/appropriate foundation models is the main assumption of the method and this needs to be expanded on.
- The method combines some of the existing ideas but the paper is written as if all the ideas originate from the authors. This should be addressed.

Related Work:
- Most of the relevant papers about foundation models and RL are cited but they are grouped together. A few of the most relevant papers can be expanded upon.
- There are "human demonstration seeded RL" papers that utilize even less amount of training. The authors touch upon this issue by mentioning the use of teleoperation. However, there are also "inverse RL" papers that can be interpreted to acquire value and success "priors" from demonstrations.
-- Eteke et al., Reward Learning From Very Few Demonstrations
-- Wu et al., SQUIRL: Robust and Efficient Learning from Video Demonstration of Long-Horizon Robotic Manipulation Tasks
- Similarly there are "date-efficient" RL methods to learn manipulation tasks but admittedly they are still not as efficient as the presented paper:
-- Popov et al., Data-efficient Deep Reinforcement Learning for Dexterous Manipulation

Method:
- RL formulation is missing. The action space, state space and discount factor are not stated.
- The assumptions and requirements on the foundation models can also be described here.
- Similar to the above point, the assumption of the performance of the "success-reward priors" is a bit shaky, as the authors do not seem to train a binary classifier but rely on the last image of the task. Note that the ground-truth is used for the simulation. The learned classification model with 50k data points, uses data from a trained policy.
- On a related note, the fact that goal configurations are needed for the value prior (ViP) is not explicitly stated.
- If the authors want to address the "prior" semantics, this is a good section to do it
- The way to acquire such priors is a core part of the paper in my opinion and describing them in the appendix leaves the paper a bit on the air. There is not much space for this but maybe the way it is done can be summarized in this section.
- I am a bit curious about the policy regularization term. The prior policy is shown to be useful in reducing the exploration steps. However, it is not very successful and as such, one would expect using this term in the long run to hinder the performance which was not the case. Is there anything that was done specifically to avoid this (e.g. removing this term after some steps/based on selection frequency, setting the loss weight etc.)
- In equation 1 second line, "y" is not specified (even though it is apparent for people working on RL)

Experiments:
- It looks like there are some hyper-parameter tuning going on for specific tasks. This is not a deal breaker but it is also hidden in the appendix.
- The noisy prior tests mainly use "symmetric" noise. The only systematic noise is for the discretized policy. What would happen if the priors are systematically wrong (e.g. wrong directions, persistent wrong success signals for certain configurations)? Moreover, is there any such evaluations for the value prior (i.e. ViP)? Due to the time and space concerns, these can be left for discussion.

Discussion:
- Limitations are written more like future directions.
- The human and fine-tuning efforts are not addressed in the limitations.

Other:
- There are grammar and spelling issues throughout the paper. This starts from the title! It should be either "Let Embodied Agents Learn on Their Own" or "Let The Embodied Agent Learn on Its Own". However, I recommend changing the title entirely.


After rebuttal:
- Most of my concerns are addressed and I have updated my scores accordingly.
- About the title, I am not going to insist on any change but my suggestion would be similar to what the authors propose: "Leveraging Foundation Priors for Sample Efficient and Reward Engineering Free Reinforcement Learning". If authors want to keep their current title, using "the embodied agent" still sounds wrong to my ear since it implies a specific embodied agent.
- There are some typos both in the original test and the new additions introduced so I strongly recommend the authors to fix them for the camera ready.
- I recommend that the authors add the prior policy loss discussion (weight decrease) that the authors presented in their rebuttal in the paper or the appendix.

**Quality Of The Limitations Section:**

3

**Questions For Rebuttal:**

Take the review as the main source. This part emphasizes some of the more important aspects:

- Concerns about the tite
- Concerns about the specific contributions of the method
- Addressing the assumptions about the foundation models. On a related note, a discussion on different types of noise for the acquired priors.
- The RL formulation should be clearly stated. The simulation one can be left for the appendix.
- How to acquire the priors are left for the appendix, is it possible to summarize them in the main text?
- How are the losses combined? What are their respective weights? The loss curves can be given in the appendix to help visualize their contribution to learning in addition to the ablations.
- Collecting the required human effort (e.g. prompt engineering, fine-tuning, providing goal states etc.) in a single place (e.g. discussion) would better serve the paper.

**Robotics Focus:**

4

**Summary Of Paper:**

The paper utilizes "foundation models" to improve the efficiency of reinforcement learning of robotic manipulation tasks. The idea is to leverage these models to get policy priors (to help with exploration), value/reward "priors" (provide dense signals, mostly for "reward shaping") and success "priors" (the final and arguably the main reward signal which is sparse). This way, there is no need to provide human demonstrations and/or to engineer rewards. The effort is shifted to prompting the foundation models (authors imply but do not explicitly state that this is easier) and designing primitive robot actions. The authors test their approach on both real world and simulated manipulation tasks. There is no strong baseline for the robotic task but the results are positive. For simulated tasks, the presented method performs much better than the regular RL algorithms with engineered rewards for the allocated time budget. The authors also ablate the components of their approach.

**Summary Of Recommendation:**

Overall, the results are strong and limitations are sufficiently presented and discussed.

---

### Author Rebuttal · Authors · 2024-08-09

The revised paper is attached in the file.

---

### Decision · Program_Chairs · 2024-09-04

**Decision:**

Accept

**Comment:**

The paper was initially evaluated with mixed but mostly positive reviews. After the rebuttal, the reviewers suggest strongly accepting the paper.

Here is a high level overview of the reviews. (before rebuttal)
### Strengths
- The paper performs a comprehensive study of policy, reward, and value priors in both simulation and real-world settings, demonstrating impressive sample efficiency and strong results. (Reviewer M4P9, Reviewer kNo7)
- The approach is sound and novel, combining the acquired "priors" effectively, and includes sufficient ablations to validate its effectiveness. (Reviewer M4P9)
- The paper shows good results and extensive experiments, evaluated on a range of real-world and simulated tasks. (Reviewer bb5a)

### Weaknesses
- The title is convoluted and not informative enough, and the introduction lacks proper historical context with some inaccuracies. (Reviewer M4P9)
- The priors, especially the policy, heavily rely on human engineering for designing low-level skills and prompts, and the method's claim of being "agnostic to foundation models" needs expansion. (Reviewer M4P9, Reviewer kNo7)
- The RL formulation is missing crucial details such as action space, state space, and discount factor, and the paper lacks generalization experiments beyond the predefined training range. (Reviewer M4P9, Reviewer kNo7)
- The storyline should be changed, as combining common techniques is not enough to call it a general framework, and additional ablations are needed for better validation. (Reviewer bb5a)

Most concerns were addressed in the rebuttal.
The authors are asked to read the updated reviews and make sure to implement the promised and suggested changes.
As an additional note:
- the font sizes in the graphics are way too small, please update them. (e.g. by using a small fig_size in matplotlib)